# Lipid Rafts Interaction of the ARID3A Transcription Factor with EZRIN and G-Actin Regulates B-Cell Receptor Signaling

**DOI:** 10.3390/diseases9010022

**Published:** 2021-03-20

**Authors:** Christian Schmidt, Laura Christian, Tyler A. Smith, Josephine Tidwell, Dongkyoon Kim, Haley O. Tucker

**Affiliations:** 1Department of Molecular Biosciences, the University of Texas at Austin, Austin, TX 78712, USA; schmidt102@gmail.com (C.S.); laura.manno@gmail.com (L.C.); tylersmith128@utexas.edu (T.A.S.); josephinetidwell@gmail.com (J.T.); kimdk1@yahoo.com (D.K.); 2Department of Biomaterials and Healthcare, Division of Life Science and Bioprocesses, Fraunhofer-Institute for Applied Polymer Research (IAP), 14476 Potsdam-Golm, Germany; 3Department of Biological Sciences, Rensselaer Polytechnic Institute, Troy, NY 12180, USA; 4Atreca, Inc., South San Francisco, CA 94080, USA

**Keywords:** transcriptional regulation, lipid rafts, actin cytoskeleton, B-cell antigen receptor

## Abstract

Several diseases originate via dysregulation of the actin cytoskeleton. The ARID3A/Bright transcription factor has also been implicated in malignancies, primarily those derived from hematopoietic lineages. Previously, we demonstrated that ARID3A shuttles between the nucleus and the plasma membrane, where it localizes within lipid rafts. There it interacts with components of the B-cell receptor (BCR) to reduce its ability to transmit downstream signaling. We demonstrate here that a direct component of ARID3A-regulated BCR signal strength is cortical actin. ARID3A interacts with actin exclusively within lipid rafts via the actin-binding protein EZRIN, which confines unstimulated BCRs within lipid rafts. BCR ligation discharges the ARID3A–EZRIN complex from lipid rafts, allowing the BCR to initiate downstream signaling events. The ARID3A–EZRIN interaction occurs almost exclusively within unpolymerized G-actin, where EZRIN interacts with the multifunctional ARID3A REKLES domain. These observations provide a mechanism by which a transcription factor directly regulates BCR signaling via linkage to the actin cytoskeleton with consequences for B-cell-related neoplasia.

## 1. Introduction

The cortical cytoskeleton is an actin-based structure that underlies and confers mechanical support to the plasma membrane (PM) via a dense meshwork (about 100 nm thick) comprised primarily of filamentous (F) actin [1]. An array of proteins, including myosin motors as well as actin-binding and linker factors, comprise the PM [2]. The cortex is dynamic in that it can locally disassemble (globular (G)-actin) and reassemble (F-actin) to drive the formation of cellular protrusions and to modulate lipid microdomain (lipid raft) organization [3,4]. PM-associated receptors maintain dynamic contacts with the cortex via linker proteins that simultaneously bind actin and the cytoplasmic domains of PM proteins [5]. The resulting receptor–cytoskeleton anchoring has a direct impact on receptor mobility and influences receptor function [6]. Over 100 actin-binding proteins constitute the cortical cytoskeleton [1].

In resting B cells, the mobility of B-cell antigen receptors (BCRs) is restricted, as the cortical actin cytoskeleton acts as a barrier to their diffusion [7,8]. Such contractile stress and tension is mediated by ERM proteins (EZRIN, RADIXIN, and MOESIN) [9,10]. ERM proteins supply the tension needed to anchor BCRs in the cortex. Actin and EZRIN form a network which confines BCRs in nanoscale lipid rafts via EZRIN carboxyl terminal binding to integral membrane proteins of the BCR complex [11]. BCR mobility, on the other hand, is established if the EZRIN–actin interaction is disrupted [12].

Besides its mechanical functions, actin has also been implicated in transcriptional regulation—both through cytoplasmic alterations in cytoskeletal actin dynamics and through the assembly of transcription factor (TF) regulatory complexes [13]. Established examples include the subcellular localization of myocardin-related TFs (MAL, MKL1, BSAC, and MRTF-B) [14,15], the PREP2 homeoprotein TF, and the TF repressor YY1 [16,17].

Lymphoma and other immunological diseases, including autoimmune disorders, can derail the normal activation of B cells via impairment of the actin cytoskeleton. These include Wiskott–Aldrich syndrome (WAS), an immunodeficiency disease resulting from deficiency of WASP or WASP-interacting protein (WIP)—both critical actin regulators in hematopoietic cells [18,19]. Diffuse large B-cell lymphoma (DLBCL) is also highly associated with elevated levels of phosphorylated actin-binding proteins, including ERMs [20]. Likewise, defects that alter CD19 and other BCR co-stimulatory receptor expression or alterations in intracellular signaling thresholds/strength can lead to disease states. For example, autoimmune diseases often derive from the flawed regulation of B-cell responses which result in the emergence of high-affinity autoreactive B cells, autoantibody production, and tissue damage (reviewed in [21]).

We observed [22] that a palmitoylated pool of ARID3A, a TF restricted to B cells/embryonic stem cells, shuttles from the nucleus to the PM where it is diverted to lipid rafts of resting B cells to associate with “signalosome” components. The BCR signalosome requires the participation of signaling proteins (e.g., BTK and BLNK) whose genetic defects often impair B-cell activation, differentiation, and often lead to agammaglobulinemia [11,12]. Following BCR stimulation, ARID3A transiently interacts with SUMOylation enzymes, blocks calcium flux and phosphorylation of BTK and TFII-I TFs, and is then discharged from lipid rafts as a SUMO-I-modified form [22]. The lipid raft concentration of ARID3A contributes to the signaling threshold of B cells, as their sensitivity to BCR stimulation decreases as the levels of ARID3A increase [22]. These events are independent of the previously established role of nuclear ARID3A in immunoglobulin gene transcription [23].

Here we demonstrate that actin is a critical component underlying ARID3A-regulated BCR signal strength. ARID3A interacts with actin exclusively within lipid rafts via the actin-binding protein EZRIN, which in unstimulated BCRs is confined within lipid rafts. We observed that antigen binding discharged the ARID3A–EZRIN complex from lipid rafts, allowing the initiation of downstream signaling events. Consistent with the timing of EZRIN discharge, ARID3A–EZRIN interaction occurred almost exclusively within unpolymerized G-actin, and was mediated by the multifunctional REKLES domain of ARID3A. Along with our previous data, this provides further mechanistic insight into how ARID3A can directly regulate BCR signaling via association with the actin cytoskeleton. Our results further suggest that ARID3A deregulation may underlie BCR-actin-associated immunological diseases.

## 2. Results

### 2.1. Variable Levels of ARID3A within Lipid Rafts in Unstimulated and Stimulated B-Cell Lymphomas

From B-cell leukemias/lymphomas, we prepared whole-cell lysates (WCLs) as previously described [22]. Lipid rafts were prepared by flotation on discontinuous sucrose gradients [22] using the B-cell-specific lipid rafts component Raflin [24] as an internal control for purity (Materials and Methods) (Appendix A). Plasma membranes (PMs), isolated as pellets from the same raft extraction, were further purified for 30 min on ice as described in the Materials and Methods. The purity of the lipid raft and plasma membrane preparations was demonstrated by the fact that neither GM130 (a member of the Golgi family) nor lamin B (a member of the nuclear envelope complex) were detectable in the preparations (Appendix A).

As shown in Appendix A, lipid rafts isolated from human transformed cell lines representative of a mature B-cell lineage (Raji, Ramos, CL-01 and Daudi) displayed detectable but consistently variable levels of ARID3A. Notably, rafts from Raji and Daudi cells (lanes 1 and 3) contained lower amounts of ARID3A compared to those from Ramos and CL-01 cells (lanes 2 and 4). Equal loading was demonstrated by probing the blots with anti-Raftlin antiserum.

Semi-quantitative densitometry (Materials and Methods) indicated that ~1% (Raji and Daudi) vs. ~10% (Ramos and CL-01) of total ARID3A partitioned within rafts, and that raft-localized ARID3A accounted for <10% of total cellular ARID3A—a value similar to the previously estimated level of membrane (m) IgM in lipid rafts [25,26]. Equal loading was demonstrated by probing the blots with anti-Raftlin. The differences among these lines in raft-localized ARID3A made them excellent choices for our analyses of BCR signaling strength.

### 2.2. BCR Stimulation Induces a Rafts-Restricted, Transient Interaction of ARID3A with EZRIN Prior to Their Co-Discharge

We previously demonstrated that Ramos and CL-01 were less sensitive, whereas Raji and Daudi were more sensitive to BCR stimulation, as judged by the downstream production of tyrosine phosphorylation and Ca^2+^ mobilization [22]. We further showed that the engagement of the BCR results in a significant and specific reduction of the small pool of lipid-rafts-localized ARID3A, which is lost from lipid rafts as BTK and other signalosome components accumulate there in proportion to BCR signaling strength [22].

The fact that the cytoskeletal linker protein EZRIN undergoes a similar anti-IgM-mediated discharge from lipid rafts [11,12] prompted us to examine ARID3A–EZRIN interaction under differential BCR signal strength conditions. As shown in Figure 1, we observed that EZRIN and ARID3A did not form an immunoprecipitable complex in WCLs of any of the B-cell lines, regardless of whether the cells were stimulated (Figure 1A(a–c), lanes 5, 6, 11, 12, 17, 18, 23, and 24). However, EZRIN and ARID3A did associate within lipid rafts, and their interaction required BCR stimulation (compare lanes 7 and 19 with lanes 8 and 20 in Figure 1A(a,b)).

EZRIN was co-discharged with ARID3A as evidenced by their identical trafficking. That is, EZRIN was detected in anti-ARID3A IPs of lipid rafts of the higher-threshold cell lines (Ramos and CL-01) stimulated with anti-IgM only (Figure 1A(a), lanes 8 and 20); within all cell lines stimulated with anti-CD19 only (Figure 1A(b), lanes 2, 8, 14, and 20); and discharged in all cell lines following anti-IgM+anti-CD19 stimulation (Figure 1A(c), lanes 2, 8, 14, and 20) (Appendix A). Notably, EZRIN was retained in the membrane fraction only following strong co-stimulation (Figure 1A(c), lanes 4, 10, 16, and 22). We observed no change in the expression levels of SUMO-1, which modifies ARID3A at a single lysine residue and interacts with ARID3A in rafts (22; Appendix A).

These results suggest that post-discharge, the ARID3A–EZRIN complex is directed to different fates depending on the signaling threshold of the BCR.

### 2.3. EZRIN Knockdown Results in Reduced BCR Signaling Strength

The above results suggest that, within lipid rafts, EZRIN might link ARID3A indirectly to the BCR to control signaling strength. Thus, we tested if signal strength was impaired following strong (anti-IgM+anti-CD19) stimulation in B cells deficient in EZRIN.

Retroviral shRNA knockdown of EZRIN was performed in each of the four lymphoma/leukemia lines. ImageJ analysis confirmed each KD to be robust (75–90% relative to a scrambled sh-RNA KD control; Appendix A). EZRIN wild-type (WT) and KD B-cell leukemias/lymphomas were then loaded with Indo-1, incubated with an Fcγ III/II receptor-blocking Ab (to prevent Ab–Ag immune complexes), and then strongly stimulated with anti-CD19+F(ab′)2-IgM 30 s rior to the measurement of Ca^2+^ flux. Cells were gated, and representative results of three separate experiments assessed via the Acapella Kinetic Intensity Analysis script (Materials and Methods) are shown in Figure 1B. EZRIN loss led to significant reduction (*p* ≤ 0.05; *) in the signaling strength of the more sensitive Raji and Daudi B cells as compared to the less-sensitive Ramos and CL-01 B cells. That is, ~38% of Raji and Daudi responded to strong BCR stimulation as compared to ~12% of Ramos and CL-01, as judged by ImageJ analysis.

These data led us to speculate that the ARID3A–EZRIN complex might function to secure the raft environment for the active BCR or to prepare inactive BCR complexes for signaling activity. They further implied that the inherent signaling strength of the BCR, in the absence of EZRIN, may contribute in a feedback manner by directing the ultimate fates of ARID3A–EZRIN complexes within the actin cytoskeleton—a hypothesis tested below.

### 2.4. ARID3A Interacts via EZRIN with Actin in Leukemic Cell Lines and in Normal Murine B Cells

Previous reports established that EZRIN binds specifically to F-actin polymers [11,12] and that partial depolymerization of actin increases the strength of BCR signaling [27]. This and the data above suggested that EZRIN mediates the BCR ligation-dependent aggregation of lipid rafts by releasing them from the underlying cortical actin cytoskeleton. As shown in Figure 2A, actin and ARID3A co-immunoprecipitated in lipid rafts of all unstimulated B-cell lines (lanes 1, 5, 9, and 13). Strong stimulation with anti-IgM+anti CD19 (lanes 4, 8, 12, and 16) discharged ARID3A from actin in the lipid rafts of all cells. Weaker stimulation with either anti-CD19 or anti-IgM discharged actin from ARID3A in the lipid rafts of B cells with high signaling strength (Raji and Daudi; lanes 2, 3, 14, and 15), but not from B cells with weak signaling strength (Ramos and CL-01; lanes 6, 7, 10, and 11). Strong stimulation (Figure 2A; lanes 4, 6, 12 and 16) was required to discharge other ARID3A-interacting proteins [22], including UBC-9, PIAS-1, and SUMO-I.

The fact that ARIA3A was detected in lipid rafts implied it was discharged after strong stimulation. However, our data do not exclude whether other unknown proteins were also discharged, as no Abs against these putative factors were employed.

Next, we employed normal murine follicular (FO) B cells to test whether ARID3A interacts with EZRIN, with actin, or with both. Further, we asked whether either or both of these interactions, as demonstrated in B-cell lymphomas (Figure 2A), were dependent upon the signaling strength of the BCR. As shown in Appendix A, equal inputs of ARID3A and EZRIN within FO WCLs were stimulated as detailed above with either α-IgM, α-CD19, or α-IgM+α-CD19 and then employed as inputs for Co-IP analyses. Anti-ARID3A immunoprecipitates were fractionated on SDS-PAGE and then blotted with either anti-EZRIN or anti-αβ-actin (Appendix A, respectively). We observed that both EZRIN and actin interacted with ARID3A most avidly following strong signaling via α-IgM+α-CD19.

Finally, support for an ARID3A-actin interaction was provided by immunofluorescence data (stains and microscopy detailed in the Materials and Methods). Strong stimulation (anti-IgM+anti-CD19) was observed in FO B cells, which were stained for actin (green), ARID3A (red), and nuclei (blue). The image in Figure 2B reveals significant levels of actin-ARID3A overlap (yellow). More definitive data was provided by the higher-resolution modulated polarization microscopy (MPM) (Appendix A). MPM measures cytoskeletal filaments based on their birefringence, and differs from standard polarization microscopy by exploiting the angle dependence of birefringence.

Collectively, the data suggest that ARID3A interacts with actin which, as previously demonstrated [11,12], interacts with EZRIN. However, the data neither identify the stoichiometry nor whether ARID3A interacts directly with actin, with EZRIN, or with both.

### 2.5. ARID3A Localizes within Monomeric G-Actin 

We reasoned that a functional consequence of ARID3A discharge is the release of lipid-rafts-associated actin, either for depolymerization or for polymerization. The degree to which release is achieved might contribute to the signaling threshold of the BCR.

In nonmuscle cells of mammals, actin is encoded by two genes, *Actb* and *Actg1*, which respectively encode beta-actin and gamma-actin [28,29,30]. Each isoform can exist as a monomer, termed globular (G)-actin, or as a polymer, filamentous (F)-actin [31,32]. Changes in actin organization are driven by the assembly and disassembly of F-actin. This dynamic turnover is regulated by a diverse set of proteins, many of which bind to F-actin to influence the network architecture [31,32].

Following transfection of ARID3A into COS7 cells, we employed Triton X-100 with or without Tween 20 fractionation to obtain detergent-soluble (enriched in G-actin) and insoluble (enriched in F-actin) proteins as described in the Materials and Methods. Tween 20 is a surfactant typically employed to decrease background staining and enhance reagent spreading in manual procedures.

Following lysate fractionation over SDS-PAGE, Western blotting was performed to determine the partitioning of ARID3A and EZRIN. Acid sphingomyelinase (SMPD1) served as a positive control for the insoluble (IS) fraction, and levels of input were adjusted using a pan-actin Ab that reacts equally with both forms. Unexpectedly, given that the vast majority of actin-binding proteins associate with F-actin [30,31], ARID3A partitioned almost exclusively within the unpolymerized, G-actin soluble fraction (Figure 3A).

These unexpected results encouraged us to analyze actin-associated partitioning of a highly conserved ARID3A paralogue, ARID3C [33]. ARID3C is significantly condensed relative to ARID3A (Appendix A), but ARID3C shares all functional domains and undergoes nuclear-cytoplasmic shuttling, with a fraction of ARID3C also localizing within lipid rafts following BCR stimulation [33]. These shared features provided the opportunity to readily and more broadly investigate essential sequences required for G-actin partitioning. Indeed, we observed ARID3C partitioning into the soluble fraction (Figure 3A).

Alternatively, co-fractionation of ARID3A and 3C within G-actin might owe, at least in part, to the fact that higher detergent concentration releases them into fully soluble protein pool independently of actin. Deletions might render the ARID proteins less soluble, rather than disrupting their association with actin. We did not experience any obvious solubility issues while working with this system, but to more conclusively rule out this possibility, we employed bacterially purified ARID3A in an in vitro reaction with purified F- and G-actin. As shown in Figure 3D, we observed virtually exclusive segregation of ARID3A within the G-actin soluble fraction.

### 2.6. SUMO Modification Does Not Control ARID3 Partitioning within G-Actin

We previously demonstrated that for BCR signaling, ARID3A required SUMOylation at a single conserved lysine residue (K402) in order to exit lipid rafts ([22]; Appendix A). ARID3C requires SUMOylation of the equivalent K284 for raft exit [33]. As shown in Figure 3B, both ARID3A-K402A and ARID3C-K284A fractionated primarily within the soluble fraction. Thus, the results suggest that SUMOylation plays no major role in G-actin association.

The REKLES-β domain is essential for ARID3A association within G-actin. The REKLES domain is named for a hexapeptide (boxed in Appendix A) that is conserved among all ARID paralogues [34,35]. ARID3C encodes an additional splice variant (termed ARID3CΔ6) which lacks the C-terminal portion of exon 6—the strongly conserved protein–protein interaction region (Appendix A). ARID3CΔ6 fails to undergo nuclear-cytoplasmic shuttling, and it does not associate with ARID3A in solution [33]. Yet, it does bind to common ARID3A DNA-binding sites in vitro [33]. As shown in Figure 3B, loss of this portion of the REKLES domain resulted in transfer of ARID3CΔ6 to the insoluble fraction.

Sequence alignments allow the REKLES domain to be divided into three sub-domains: a conserved 17 amino acid N-terminal REKLES-α, a relatively conserved 51 amino acid spacer region, and a highly conserved 59 amino acid C-terminal REKLES-β domain (Appendix A). REKLES-α is required for nuclear localization (NLS) and REKLES-β is essential for nuclear export (NES) [22,33]. REKLES-β is also essential for ARID3A and ARID3C heteromeric interactions [35].

The ARID3CΔ6 observation suggested that the corresponding region of ARID3A, the REKLES-β domain, is critical for ARID3A residence within G-actin. To test this, we first determined if ARID3A carrying a complete deletion (d) (amino acids 500–562; Appendix A) of the 63 residue REKLES-β was also transferred to the soluble G-actin fraction. Accordingly, only ~40% of REKLES β-deficient d500–562 remained within the G-actin fraction. REKLES-β amino acids 521–541 contain a cluster of residues (G532, Y535, G537, and L539) conserved in all ARID3 paralogues (Appendix A; [27]). Alanine substitutions within each abolished G-actin soluble fraction association to varying degrees.

### 2.7. The REKLES-β Domain Is Essential for EZRIN–ARID3A Interaction

The data above suggested that ARID3A interaction within PM G-actin is mediated through its REKLES-β subdomain, potentially via interaction with the cytoskeletal linker protein EZRIN. This hypothesis was tested by the Co-IP experiments of Figure 4. We transfected COS7 cells with full-length ARID3A (residues 1–601) or with each of the following REKLES mutations: d453–500, in which the entire REKLES domain is deleted (d); d500–521, in which the C-terminal 21 residues of the spacer, located just N-terminal to the REKLES-β domain, are deleted; t1–541, in which ARID3A is truncated just N-terminal to REKLES-β; and d521–541, in which the REKLES-β domain is deleted (Figure 4A).

Transfected lysates were immunoprecipitated with a polyclonal anti-ARID3A Ab which was previously shown to pull down each of the input mutants [34,35,36]. After adjusting for equivalent levels of immunoprecipitated ARID3A inputs (Figure 4B, middle panel), WT and mutant lysates were fractionated on SDS-PAGE and then Western blotted with an EZRIN monoclonal antibody (mAb). As shown in Figure 4B, EZRIN–ARID3A interaction was abrogated only by deletion of the REKLES-β domain and N-terminal proximal section of the spacer region. Collectively these results indicate that ARID3A is tethered to G-actin via interaction of its RECKLES-β and proximal spacer domain with EZRIN.

## 3. Discussion

To our knowledge, ARID3A was the first TF shown to function in lipid rafts [22], but its residence there is not unprecedented. Both STAT1 and STAT3 [37] as well as an isoform of the IgH coactivator OCA-B [38,39] localize to lipid rafts. Several other TFs, including TFII-I, form cytoplasmic complexes [14,15,16,17]. That TFs function directly or as intermediates in receptor signaling is supported by the association of p35-OcaB with galectin-1 [40] and, in this report, ARID3A with EZRIN.

Previous reports suggested that EZRIN binds specifically to F-actin, which in turn increases BCR signaling strength by aggregating and releasing lipid rafts from the cortical actin cytoskeleton [11,12]. Here we extend these results with the observation that an ARID3A–EZRIN–G-actin complex occurred exclusively within lipid rafts which was co-discharged from rafts following BCR stimulation (Figure 1, Figure 2, and Appendix A). Our data further indicate that, within lipid rafts, EZRIN links ARID3A indirectly to the cytoskeleton to control BCR signaling strength (Figure 1 and Figure 2).

We observed significant but relatively subtle difference in Ca^2+^ fluxes among strong and weak signaling lines following EZRIN KD (Figure 1B). Follow-up experiments employing an NFAT-driven luciferase reporter assay might provide a more sensitive means of quantifying the impact of the EZRIN loss on Ca^2+^ signaling. In addition, it would be useful to knock out ARID3A in each of the B-cell lines (using CRISPR/Cas9) and to examine the effect on BCR signaling via Ca^2+^ mobilization and time courses of whole-cell protein tyrosine phosphorylation as readouts. ARID3A mutant constructs (e.g., EZRIN non-binding or lipid raft non-localizing) could then be transfected into the KO cells to investigate how EZRIN binding and lipid raft localization impact on ARID3A’s fine-tuning of BCR signaling.

We observed that ARID3A associated almost exclusively with unpolymerized G-actin (Figure 3A,B). This was unanticipated, as the majority of actin-binding proteins analyzed in this manner in the literature associate with F-actin [30,31]. The interaction requires a patch of five residues conserved among ARID3 paralogues within the 59-residue carboxyl terminal (β subregion) of the multifunctional REKLES domain (Appendix A). In addition, REKLES-β, as well as the C-terminal 21 residues of the REKLES-αβ spacer, are required to link ARID3A to EZRIN (Figure 4B). We suggest that future microscopic investigation of these interactions in leukemic cells would prove beneficial, as the majority of the data in the manuscript derive from these tumors.

Our results add another layer of complexity to the ARID3A REKLES domain. The REKLES-α subregion encodes an NLS and a binding site for SUMO-I, whereas its β subregion provides essential residues for nuclear export as well as for heteromeric interactions with EZRIN, BTK, PIAS-1, and UBC9 [22,34,35]. The three-dimensional structure of the DNA-binding domain of ARID3A has been solved [41,42], yet the structure of REKLES remains to be determined. The observed extent of REKLES’ multifunctionality as well as the architecture of the platform tethering G-actin to the REKLES–EZRIN complex is a future goal of our research.

We combined several of these features into the model in Figure 4C. We suggest that a functional consequence of ARID3A discharge from lipid rafts is the release of lipid-rafts-associated G-actin for polymerization to F-actin. The extent to which this release is achieved contributes to the signaling threshold of the BCR. Conversely, reassembly of the polymerized F-actin cytoskeleton might act to stabilize interactions among the BCR and additional signaling molecules via trapping the lipid-raft-localized signaling complex. In support of this, Total Internal Reflection Fluorescent Microscopy TIRFM studies (capable of observing the BCR, actin, and EZRIN simultaneously) revealed EZRIN–actin networks in resting B cells [43]. That Ag–Ab binding induces transient EZRIN dephosphorylation followed by detachment of lipid rafts from the actin cytoskeleton [39] is consistent with our observations (Figure 1 and Figure 2) in respect to the promotion of BCR–lipid raft interactions. However, the stoichiometry and formation kinetics of the ARID3A–EZRIN–actin complex remains to be determined.

We found it interesting in this regard that ARID3A and EZRIN share a binding partner, S100P. Dimeric S100P binds to and activates EZRIN by unmasking its F-actin binding sites [44], whereas monomeric S100P represses the DNA-binding and transactivation activity of ARID3A [36].

We feel that further experimental refinement of our model is justified from a health perspective. ARID3A dysregulation is implicated in several malignancies—primarily those derived from hematopoietic lineage cells [45]. For example, diffuse large B-cell lymphoma (DLBCL) presents both as activated B-cell (ABC) and germinal center B-cell (GC) subtypes, with the former being far more penetrant [46]. Both ARID3A and phosphorylated actin-binding proteins, including ERMs, are highly expressed in ABC-DLBCL [27,46]. While far less is known about ARID3C [47], RNA-Seq analyses recently identified it within a signaling complex consisting of protein tyrosine phosphatase receptor type R (PTPRR), α-catenin, β-catenin, and E-cadherin that form exclusively in ovarian cancer [48]. Several members of the actin-nucleating WAVE complex, including BRK1, BRICK1, and SCAR/WAVE, are direct, upregulated transcriptional targets of ARID3A, as determined by the ENCODE ChIP-seq project [49]. WAVE proteins play diverse roles, ranging from BCR activation [50,51,52] to cancer invasion and metastasis [53,54]. ARID3A direct transactivation of PFN1, a member of the profilin family of small actin-binding proteins, is an essential regulator of actin polymerization in response to BCR ligation [55]. PFN1 has also been characterized as a tumor-suppressor in human breast cancer [56].

These data suggest that modulation of EZRIN, ARID3A, and potentially ARID3C may provide both prognostic and therapeutic options for these and other malignancies.

## 4. Materials and Methods

### 4.1. Cell Lines

Raji (EBV+, Cat. No. CCL-86), Daudi (EBV+; Cat. No. CCL-213), and Ramos (EBV−; Cat. No. CSC-C1015) [57,58,59] were obtained from ATCC (Manassas, VA, USA); CL-01 (EBV−; Cat. No. NBP1-49595) [60] was obtained from Novus Biologicals, LLC (Centennial, CO, USA). Each was maintained as previously described [22]. Cells were grown and maintained in either Dulbecco’s modified Eagle medium (DMEM) supplemented with 10% fetal bovine serum (FBS; Invitrogen, Waltham, MA, USA) or in RPMI medium containing 10% FBS. CD43- B cells were prepared by negative selection of whole human blood (Gulf Coast Regional Blood Center, Houston, TX, USA) or from 10-wk BALB/c murine splenocytes.

### 4.2. Constructs 

Mutant forms of ARID3A and ARID3C used in this study were generated previously by site-directed mutagenesis [22,33,34,35].

### 4.3. Preparation of Stable sh-RNA Retrovirally Transduced B-Cell Lines 

Stable transductants were established by employment of the Phoenix-A retroviral system. Approximately 3 × 10^5^ amphitropic Phoenix-A packaging cells in 4 mL of DMEM were supplemented with 10% fetal bovine serum (FBS) in 60 mm plates. After one day of culture, cells were transfected using FuGene6, and viral supernatant was harvested 2 days post-transfection, centrifuged, and filtered to remove live cells and debris. The target cell lines described above were plated (~3 × 10^5^) onto 60-mm plates and growth medium was replaced with viral mixture. To introduce EZRIN-knockdown sequences, we used oligos ordered from Integrated DNA Technologies with restriction site overhangs BbsI and XhoI (sense, 5′-ACCGGCCGTGGAGAGAGAGAAAGATTCAAGAGATCTTTCTCTCTCTCCACGGCT TTTTTACCGGTC-3′; anti-sense, 5′-TCGAGACCGGTAAAAAAGCCGTGGAGAGAGAGAAAGATCTCTTGAATCTTTCTCTCTCTCCACGGC-3′). Stable lines were established by selection with 2 μg/mL of puromycin from day 2 post-infection.

### 4.4. B-Cell Stimulation

To measure signaling effects at low doses of anti-IgM where receptor internalization is minimized, we used monoclonal anti-IgM Abs in the absence of secondary crosslinking. To stimulate B cells, 500 ng of F(ab’)2 fragments of α-μ (clone JDC-15; Dako (α-human); OB1022; Southern Biotech (α-mouse)) and α-CD19 (clone HD37; Dako (α-human); clone SJ25-C1 (α-mouse)) were added to ~5 × 10^8^ cells for 5 min at 37 °C. We determined by FACS analysis and semi-quantitative Western blotting of lipid-raft-associated membrane (m)IgM (Appendix A; [22]) that under these conditions 1–5% of mIgM in rafts and membranes are engaged.

### 4.5. Preparation of Lipid Rafts

Approximately 500 mg of wet cell pellet was washed twice in ice-cold phosphate-buffered saline (PBS) and homogenized in 5 mL of 10 mM Tris/Cl (pH 7.4), 1 mM EDTA, 250 mM sucrose, 1 mM phenylmethylsulfonyl fluoride, and 1 μg/mL leupeptin (all from Sigma, St Louis, MO, USA) in a tightly fitted Dounce homogenizer using five strokes. The resulting homogenate was centrifuged at 900× *g* for 10 min at 4 °C, and the supernatant was then subjected to centrifugation at 110,000× *g* for 90 min at 4 °C. The resulting membrane pellet was resuspended in ice-cold 500 μL TNE buffer (10 mM Tris/Cl (pH 7.4), 150 mM NaCl, 5 mM EDTA, 1% Triton X-100 (Sigma), 10× protease inhibitors (Complete tablets, Roche, Indianapolis, IN, USA)). Sucrose gradients for the preparation of lipid rafts were assembled as previously described [18]. Lipid rafts were isolated by flotation on discontinuous sucrose gradients. Membrane pellets were extracted for 30 min on ice in TNE buffer. For the discontinuous sucrose gradient, 1 mL of cleared supernatant was mixed with 1 mL of 85% sucrose in TNE and transferred to the bottom of an ultracentrifugation tube, followed by overlay with 6 mL of 35% sucrose in TNE and 3.5 mL of 5% sucrose in TNE. Samples were spun at 200,000× *g* for 30 h at 4 °C. Fractions were collected from the top of the gradient and analyzed using Western blotting and/or coimmunoprecipitation, as described [30,31].

### 4.6. Immunoprecipitation/Western Blot Analyses

We employed a stringent RIPA formulation of 500 mM NaCl; 10 mM Tris–HCl, pH 8; 0.1% SDS; 5 mM EDTA, pH 8; 10× protease inhibitor (Complete tablet, Roche) to solubilize lipid rafts for subsequent immunoprecipitation experiments. Briefly, buoyant fractions, taken from the discontinuous gradient centrifugation, were pooled and incubated with the same volume of RIPA buffer on ice for 15 min. Resulting extracts were pre-cleared by rocking with 1 mL of a 5% slurry of RIPA equilibrated Protein A beads (CL-4B; Amersham Pharmacia, Uppsala, Sweden) for 4 h at 4 °C. After removal of the precipitate, the resulting supernatant was subjected to IP/Western blot assays. The following Abs were employed: α-CD19 (clone 6D5, Dako), α-ARID3A polyclonal Ab (produced in-house; [19]), α-IgM (BD Pharmingen), α-V5 (Sigma), α-Raftlin (graciously provided by Dr. Akihiko Yoshimura; 21), α-SUMO-1 (Sigma), anti-TFII-I (kindly provided by Dr. Carol Webb; [18]), pan α-actin (rabbit origin, Cytoskeleton, Inc.), α-Acid sphingomyelinase (SMPD1; ab83354, Abcam), goat α-EZRIN (sc-6407; Santa Cruz)

### 4.7. Accumulation of Cytosolic Calcium

Indo-1 AM (acetoxymethyl ester; Invitrogen) was added to ~3 × 10^6^ leukemia/lymphoma cells in 500 μL Hank’s Balanced Salt Solution (HBSS; Invitrogen) and 10% FCS (HBSS-10). The final Indo-1 concentration was 1 μM. Following incubation at 37 °C for 30 min, cells were kept at RT for 5 min and then washed with HBSS containing 2 mM HEPES buffer and no serum (HBSH-0). All manipulations including the incubation were in the dark. Cells were then reacted with a rat anti-human CD16/CD32 Fc blocking Ab (clone 2.4G2, Pharmingen) and washed with HBSH-0 in the dark at 4 °C. The cells were resuspended in 50 μL HBSH-0, added to a total volume of 150 μL. Prior to Ca^2+^ analyses, cells were filtered at RT, warmed to 37 °C, and then placed in a flow cytometer maintained at 37 °C at a flow rate of 250 cells/s. Anti-IgM and anti-CD19 were added (detailed as above in Section 4.4) 30 s prior to initiation of the experiments. Data acquisition employed a 30 s baseline and was continued for 300 s at 37 °C. Cells were analyzed at ~250 cells/s, and their flux in calcium concentration was determined as the 485/405 nm emission ratio with excitation at 355 nm. Calibration was performed by measuring Rmin and Rmax, and applying the equation described previously [48] using the Acapella Kinetic Intensity Analysis script. Responses are reported as [Ca^2+^] vs. time.

### 4.8. Calcium Calibration

The presence of unhydrolyzed dye might affect calibration significantly. The 485/405 nm fluorescence ratios were converted into calcium concentrations using the analytical expression: [Ca^2+^] = Kd × B × (R − Rmin)/(Rmax − R) in which Rmax and Rmin are determined from the measurement. At the end of the measurement the intracellular dye was saturated with calcium. By using a Ca^2+^ ionophore and saturating [Ca^2+^], the Ca^2+^-insensitive AM form will not undergo a wavelength shift while the sensitive form will.

### 4.9. Purification of Actin

To obtain detergent-soluble (actin-free) and insoluble (actin-enriched) fractions, ARID3A-transfected COS7 cells were harvested by scraping, and then resuspended in 750 μL of pre-warmed LAS buffer (50 mM Pipes (pH 6.9), 50 mM NaCl, 5 mM MgCl_2_, 5 mM EGTA, 5% (*v*/*v*) glycerol, 0.1% NP-40, 0.1% Triton X-100 (with or without 0.1% Tween 20), 0.1% β-mercaptoethanol, 1 mM ATP, 0.001% Antifoam C, and a protease inhibitor cocktail consisting of 0.4 mM tosyl arginine methyl ester, 1.5 mM leupeptin, 1 mM pepstatin A, and 1 mM benzamidine). Cells were lysed by 10 passages through a 25-gauge needle. The lysate was clarified by centrifugation (400× *g* for 5 min at RT). The supernatant (100 μL) was collected, and actin polymerization was initiated by addition of 2 mM MgCl_2_, 0.2 mM EGTA, 100 mM KCl, and 1 μM phalloidin and by incubation for 1 h at 37 °C. F-actin and F-actin-binding proteins were pelleted by ultracentrifugation (270,000× *g*) for 1 h at 4 °C. The supernatant (G-actin fraction) was collected, and the pellet (F-actin fraction) was washed twice with Milli-Q water and resuspended in 100 μL of 8 M urea. After addition of SDS loading buffer, the samples were separated by SDS-PAGE and analyzed by Western blotting for actin using anti-pan-actin polyclonal rabbit Ab (1:2000; Cell Signaling Technology, 4968).

### 4.10. Immunofluorescence Staining of Murine B Cells

Mouse CB7Bl/6 MZ and FO B cells were isolated from mouse spleen cell suspension by anti-CD19 exclusion using a kit #130-100-366 (Miltenyi Biotec, Cologne, Germany). Cells were fluorescently stained with rabbit αβ-actin mAb (SP124; ab115777; Abcam) developed with goat anti-rabbit IgG H&L (Alexa Fluor^®^ 594), mouse anti-ARID3A mAb A-4 (sc-398367, Santa Cruz Biotechnology) developed with goat anti-mouse IgG H&L (Alexa Fluor^®^ 488; ab150117, Abcam), and stained with DAPI (Staining Solution ab228549, Abcam). Visualization is described below and shown in Figure 2B and Appendix A.

### 4.11. Quantification of Blot Intensities with ImageJ

Quantitative results of Western blots were assessed using ImageJ, a Java-based image analysis package downloaded from the National Institutes for Health at https://imagej.nih.gov/ij/download.html (access date 21 May 2014) upgraded to the latest full distribution (including macros, plugins and LUTs) as detailed at http://wsr.imagej.net/distros/ (access date 23 June 2015). To determine the dynamic range and limit of detection, blots were wrapped in clear plastic for 60 and 120 s. For X-ray film exposure, blots were placed in a cassette and exposed to film for 20, 60, and 300 s. The developed film was scanned using a Molecular Image GS-800 densitometer (Biorad). Data was analyzed using the Volume Analysis Specification Quantity in Quantity One software (FPQuest™).

### 4.12. Modulated Polarization Microscopy (MPM)

Modulated polarization microscopy (MPM) visualizes cytoskeletal filaments based on their birefringence, but differs from standard polarization microscopy by exploiting the angle dependence of birefringence. MPM imaging as detailed by Kuhn and Poenie [61,62] was performed at a rate of one to two processed frames per second. Each image in the resulting MPM movie sequence was enhanced slightly using a 3 × 3 convolution kernel of [(−1/2 1/2 ½), (−1 1/2 ½), (−1/2 1/2 ½)], which added a small emboss effect to the image and improved visibility. Fluorescent images were acquired using a 12 bit CCD camera (Model DVC-1312M, DVC, Austin, TX, USA) on a Nikon Diaphot 200 fluorescence microscope. Image stacks were obtained using a MAC 2000 *z*-axis focus controller (Ludl Electronic Products, Hawthorne, NY, USA) and a custom image acquisition plugin written for ImageJ. The point-spread function (PSF) was measured under similar conditions using 100 nm fluorescent beads (L-5473, Molecular Probes) diluted in dH2O and dried on glass slides to give approximately one bead per field. Image stacks were deconvolved for 500 iterations using the expectation maximization algorithm in XCOSM [62]. 3D projections were calculated using the maximum value projection method in ImageJ, and stereo pairs were generated using images that differed in rotation by 10 degrees. Images were enhanced using a multiscale 3D line filter implemented as a custom plugin for ImageJ.

### 4.13. Purification of ARID3A in Bacteria

An N-terminal 6X-histidine-tagged ARID3A was constructed as previously described [63]. Briefly, full-length ARID3A was cloned into the pET30a+ expression vector (Novagen), and its expression induced with IPTG 30 min after chaperone induction. Harvested cells were disrupted by sonication, and total cell lysates were analyzed on 12% SDS-PAGE (prior to or as a monitor of purification) with Silver Stain (Invitrogen). Following elimination of cell debris by centrifugation, supernatants were purified by affinity chromatography over Ni^2+^-NTA agarose (SuperFlow, ThermoScientific, Rockville, IL) according to the manufacturer’s instructions (Novagen Inc, Madison, WI and Qiagen, Germantown, MD, USA). Further purification was carried out by DEAE Bio-Gel agarose chromatography as instructed by the vender (Pharmacia Fine Chemicals, Uppsala, Sweden). The final yield of purified ARID3A was ~20 μg from LB media.

## Figures and Tables

**Figure 1 diseases-09-00022-f001:**
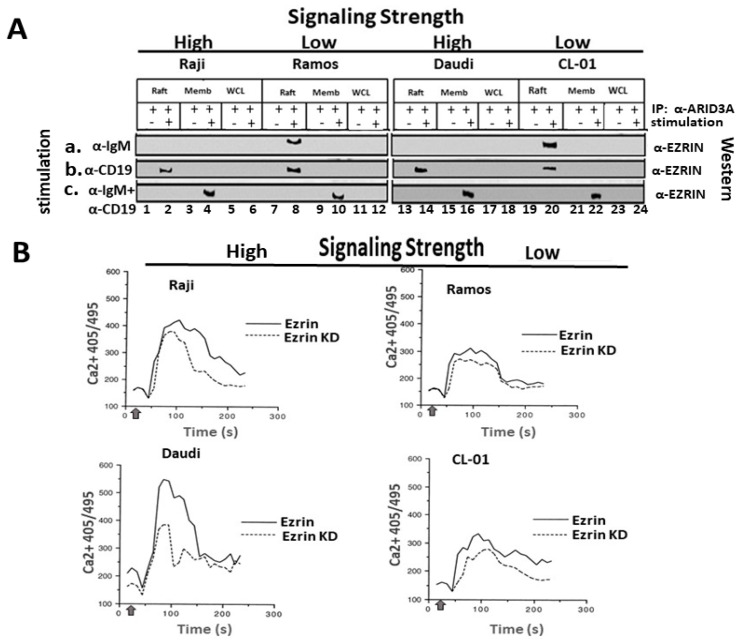
(**A**). The ARID3A–EZRIN complex is directed to different fates depending on the signaling threshold of the B-cell receptors (BCRs). Raji, Ramos, CL-01, and Daudi cells (~10^8^ cells/dish) were stimulated for 5 min with (**a**) 100 ng anti-IgM(α-IgM), (**b**) 100 ng α-CD19, or (**c**) 100 ng α-IgM +100 ng α-CD19. Lipid rafts (Raft), plasma membranes (Memb), and whole-cell lysates (WCLs) were analyzed following anti-ARID3A IP and Western blotting with anti-EZRIN. EZRIN–ARID3A did not co-IP in either stimulated or unstimulated WLCs (lanes 5, 6, 11, 12, 17, 18, 23, and 24) but did within BCR-stimulated lipid rafts (lanes 8 and 20 (panels **a** and **c**). ARID3A–EZRIN co-immunoprecipitated in lipid rafts of the higher threshold cell lines (Ramos and CL-01) stimulated with anti-IgM only (**a**, lanes 8 and 20); within all cell lines stimulated with anti-CD19 only (**b**, lanes 2, 8, 14, and 20); and were discharged in all cell lines following anti-IgM+anti-CD19 stimulation (**c**, lanes 2, 8, 14, and 20). EZRIN was retained in the PM only following strong co-stimulation (**c**, lanes 4, 10, 16, and 22). (**B**). EZRIN knockdown (KD) resulted in reduced signaling strength. EZRIN levels were reduced by shRNA knockdown (KD; Materials and Methods), and calcium (Ca^2+^) flux was measured in the indicated B-cell tumors of high or low signaling strength following strong costimulation, as detailed in Materials and Methods. Data shown are representative of three independent experiments. α-IgM and α-CD19 were added 30 s (upward arrow) before the beginning of the experiment, and the cells were analyzed at 250 cells/s. Results are plotted as the mean calcium concentration vs. time. A significant difference (*p* ≤ 0.05 *) was determined among high- vs. low-signaling cells when comparing their calcium response to stimulation with and without EZRIN knockdown.

**Figure 2 diseases-09-00022-f002:**
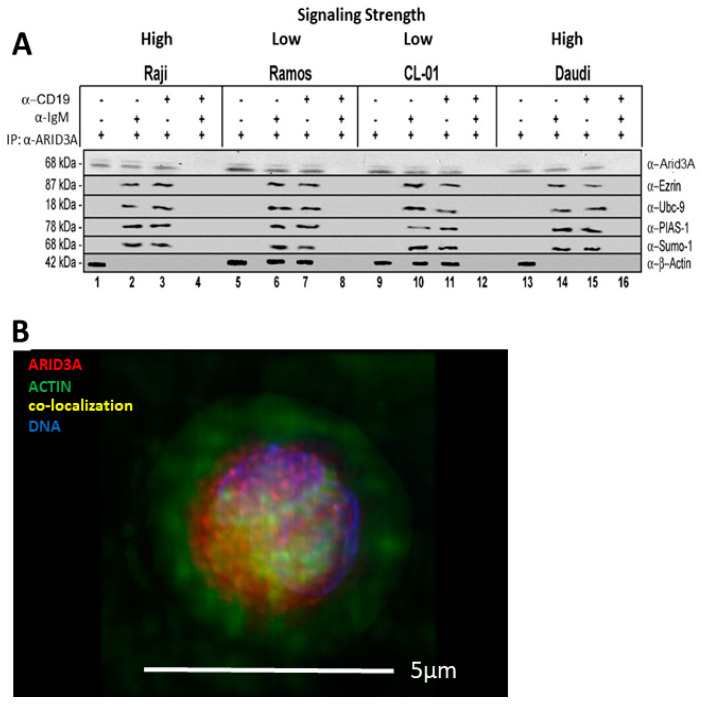
Interaction and discharge of ARID3A-actin complexes within lipid rafts as a function of BCR signal strength. (**A**) ARID3A co-immunoprecipitated with αβ-actin in the lipid rafts of B-cell lines (lanes 1, 5, 9, and 13) regardless of their signaling strength. Strong stimulation (α-IgM+α-CD19; lanes 4, 8, 12, and 16) discharged ARID3A from actin in the lipid rafts of all cells, whereas weaker stimulation (α-CD19 or α-IgM) discharged actin from ARID3A in lipid rafts of strong (Raji and Daudi; lanes 2, 3, 14, and 15), but not weak (Ramos and CL-01; lanes 6, 7, 10, and 11) signaling strength B cells. Strong stimulation (lanes 4, 8, 12, and 16) was required to discharge other ARID3A-interacting proteins (UBC-9, PIAS-1, and SUMOI) from lipid rafts. (**B**) Triple immunofluorescence of murine B cells stained for actin (green), ARID3A (red), and nuclei (blue) revealed strong actin-ARID3A association (yellow; also illustrated in Appendix A). Scale bar, 5 µm.

**Figure 3 diseases-09-00022-f003:**
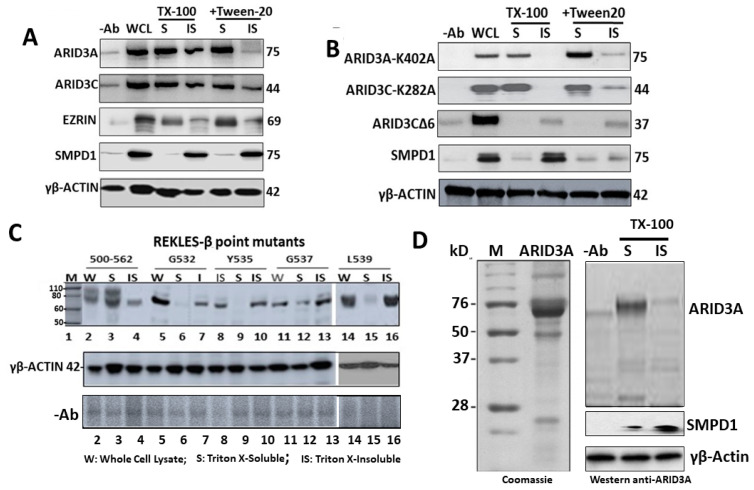
ARID3A and the related paralogue ARID3C localize within monomeric globular (G)-actin via the conserved REKLES-β domain. (**A**) ARID3A and 3C transfected COS7 lysates were processed via Triton X-100 and/or TritonX-100+Tween 20 fractionation into insoluble (IS, filamentous (F)-actin) or soluble (S, G-actin) proteins. After fractionation by SDS-PAGE, samples were Western blotted with the indicated Abs; anti-SMPD1 served as an IS positive control and actin inputs were equalized with pan-actin. ARID3A and 3C partitioned primarily within the G-actin fraction. (**B**) Loss of the REKLES domain (ARID3A-d(521541) and ARID3CΔ6), but not SUMO-I binding motifs (ARID3A-K402A or ARID3C-K284A) abolished partitioning into the soluble G-actin fraction. (**C**) A region within REKLES-β (amino acids 532–540 and indicated on Appendix A) contains a cluster of conserved residues (G532, Y535, G537, and L539) whose point mutation significantly reduced or abolished ARID3A association with G-actin. (**D**) Bacterially purified ARID3A segregated within soluble G-actin. Left: SDS-PAGE Coomassie-stained image of bacterial ARID3A following purification as detailed in the Materials and Methods. Right: SDS-PAGE Western blot of TX-100 soluble (S) and insoluble (IS) fractions blotted with the indicated probes. SMPD1, marker for the insoluble fraction; γβ-actin, loading control.

**Figure 4 diseases-09-00022-f004:**
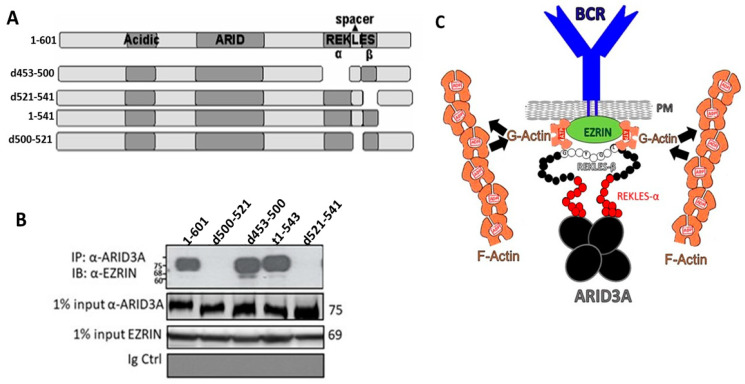
REKLES-β deletions abrogate ARID3A–EZRIN interaction. (**A**) Diagram of ARID3A REKLES deletion (d) mutants. Deletions are indicated by gaps. (**B**) EZRIN–ARID3A interaction was abrogated only by deletion (d453–500 and d521–541) of the REKLES-β domain. Full-length ARID3A and its deletion mutants of (**A**) were transfected into COS7 cells and lysates were immunoprecipitated with anti-ARID3A prior to resolution on SDS-PAGE followed by anti-EZRIN Western blotting (upper panel); normalization of equivalent inputs (middle panels); anti-EZRIN background control (lower panel). (**C**) ARID3A is tethered to G-actin via interaction of its REKLES-β domain with EZRIN. Cartoon summarizing and extending the findings of this report. We speculate that when the ARID3A tetramer (large black spheres) interacts with EZRIN (green sphere) via a patch of conserved REKLES-β residues (small black spheres with single amino acid codes in white) and proximal spacer residues (small red spheres), EZRIN is discharged from lipid rafts in order to release lipid-rafts-associated ATP-G-actin (brown semicircles), possibly for polymerization to ADP-F-actin (brown chains).

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
