# Peer review of "Lipid Rafts Interaction of the ARID3A Transcription Factor with EZRIN and G-Actin Regulates B-Cell Receptor Signaling"

_diseases, 2021, doi:10.3390/diseases9010022_

Round 1

Reviewer 1 Report

Regarding the name of proteins (all “in caps”), I recommend the authors check with the editorial which nomenclature to use.

I appreciate the authors have answered most of my comments. However, it is impossible to evaluate the present version since the Supplementary material was not provided. I will send my comments as soon as the supplementary material becomes available.

Author Response

We thank the Reviewer for his/her comments. Putting protein abbreviations in all caps is the current nomenclature for mammalian proteins as documented in the International Protein Nomenclature Guidelines of the NBCI. However, Reviewer 2 demanded that Actin be written as lower case actin unless it began a sentence, and then Actin.

Re: the supplemental file: we aplogize for its exclusion, and it is uploaded here for your review

Reviewer 2 Report

In my opinion, the first version of this manuscript was very well. I find a new version with more details and the text edited according to reviewers' comments highly beneficial. I would recommend this manuscript for publication in the present state.

Author Response

We thank Reviewer 2 for his positive remarks, which include no additional changes in the submitted draft.

Reviewer 3 Report

Despite the additional experiments performed by the authors to address my original comments, they failed to produce the key data that I requested. Instead, they state their disagreement with many of my comments. Based on this, I cannot recommend publication.

As a side note, due to recent changes in rigor and reproducibility standards, most journals do not allow references to data not shown. The authors should remove these references or show the data. Their attempt to argue with this point by leaving one incidence of data not shown is troubling.

Author Response

We strongly disagree with Reviewer 3's contention that we failed to address his criticisms. What criticisms does he/she mean? He/she does not state any particular examples. We submitted new data, we made major changes in the manuscript, and addressed each and every comment of Reviewer 3. We upload our responses to Reviewer 3's review again below, as we have been given no direction in which to make further alterations.

In addition, Reviewer 3 recommended we remove all examples of "data not shown".  We made this correction in the previously submitted revision. It is clearly stated in the our response to Reviewer 3 that the 3 cases of "data not shown" were each removed. We did not argue at all with this change as implied by this Reviewer. He/she claims that one has been intentionally left in; that is incorrect.

Round 2

Reviewer 3 Report

acceptable

This manuscript is a resubmission of an earlier submission. The following is a list of the peer review reports and author responses from that submission.

Round 1

Reviewer 1 Report

This paper addresses an interesting topic related to B-cell signaling, lipid rafts, and association of signaling components with the actin cytoskeleton. I was initially interested in this study because the conclusions seemed important and interesting. However, I feel that the data in most cases fall short of these conclusions and does not in fact support the claims of the paper. Many more experiments are needed to bring this study into a publishable form. I listed my specific concerns below.

  1. I may be missing something, but from the evidence presented it appears that ezrin and ARID3A may not associate directly – and likely don’t, given that they don’t coprecipitate. They are both found within lipid rafts after stimulation, but this by itself does not prove that they interact with each other. The authors repeatedly refer to them as interacting partners and a complex, but I don’t think there is sufficient evidence for that. Moreover, given the evidence that ARID3A coimmunoprecipitates with actin, it is likely that ezrin is recruited by actin, not the other way around.
  2. Colocalization shown in Fig 2 is not convincing at all. According to the image, actin is found mostly outside the nucleus and ARID3A mostly inside the nucleus (as expected for their major functions). Colocalization seen over the nucleus very likely results from the signal overlap of the actin on top of the nucleus and the ARID3A/DNA inside the nucleus. No amount of data processing should enable any of the more detailed conclusion from these types of images. A much higher resolution microscopy is required to provide convincing evidence of this kind.
  3. The result that ARID3A cofractionates with G-actin can be explained also by the fact that higher detergent concentration releases it into the fully soluble protein pool independently of actin. Deletion of aa 500-562 might simply make it less soluble, rather than disrupt its association with actin. If the authors want to claim that G- and not F-actin binds ARID3A they should perform in vitro binding experiments with purified proteins to confirm this. With all this, I am not convinced that ARID3A does in fact associate with actin or ezrin, directly or indirectly.
  4. I am troubled by the citations of data not shown. The authors should either show the data or not refer to it in the manuscript.
  5. I am not sure what exact detection the authors used for Western blots, they only cite ImageJ as quantification tool. I assume they use ECL detection and gray level densitometry, which are not sufficiently quantitative. The bands shown appear overly contrasted. The authors should provide these details. They should also show full non-cropped unprocessed images of all Westerns, possibly as supplemental figures.
  6. On a more minor side, there is a number of spelling and grammar mistakes. A recurring one: "actin" should not be capitalized, unless at the start of the sentence. Also, in one place, the authors refer to G-actin and G-protein, which is a completely different protein. This should be corrected.

Reviewer 2 Report

The Manuscript entitled “Lipid rafts interaction of the ARID3A transcription factor with 2 EZRIN and G-Actin regulates B cell receptor signaling” demonstrates that ARID3A interacts with Actin only within lipid rafts via EZRIN, Actin binding protein. Further, EZRIN restricts unstimulated BCRs within lipid rafts of the plasma membrane. Ligation of BCR releases the ARD3A-EZRIN complex from lipid rafts and let BCR initiate downstream signaling events.

The Manuscript is overall very well written, clear and concise.

This topic is actual and very important because of the connection of ARID3A with hematopoietic malignancies development.

In myhumble opinion, it was written very well. Experiments were conducted properly,while the Discussion is grounded and supported by facts from the literature. I would recommend this Manuscript for publication in the present form.

Reviewer 3 Report

In the article entitled “Lipid rafts interaction of the ARID3A transcription factor with EZRIN and G-Actin regulates B cell receptor signaling”,  Schmidt et al. mostly used western blot analyses to demonstrate that ARID3A (Rekles domain) interacts with actin (mostly G-actin) via Ezrin in lipid rafts, confining unstimulated BCRs. Moreover, BCR stimulation discharges the protein complex from rafts and allows the initiation of signaling events. Overall, I believe the manuscript needs a major revision before being accepted for publication. My comments are listed below.

1- This reviewer did not understand why some proteins are always in capital letters. Example: Actin, EZERIN, RADIXIN, MOESIN among others. It would be great to change that throughout the manuscript.  

2- The titles for the results and materials and methods sections are completely mixed with the manuscript text. It would be great to highlight them.

3- Lines 95-96: Please check for excess parentheses.

4- More details on quantifications of western blottings are needed. It is not explained in materials and methods how they were performed. In addition, it would be interesting to present plots showing the mean fold-changes (compared to control) and their respective error bars, for the different conditions. Error bars should take into account the independent experiments. This quantification should be added for all figures that have fold-change comparisons.

5- Legend for Figure 1: it is necessary to indicate where the legend for Figure 1A starts.

6- Line 137: please correct the p-value.

7- Line 152, data not shown: it would be interesting to add this data as supplementary material.

8- Please check all the Ca2+. They should be written properly as Ca2+.

9- Figure 2B needs a scale bar. Moreover, why did you choose not to use one of the cell lineages for the immunofluorescence images? Finally, this reviewer had difficulty seeing the colocalization points. It would be interesting to perform a more accurate analysis, using, for example, an ImageJ-based colocalization plugin.

10- Lines 448-450: This calibration procedure was not clear to me. Please clarify.

11- Line 479: Multiphoton microscopy? I wonder if a multiphoton or a conventional fluorescence microscope was used. Please, verify. Also, reference 61 does not use multiphoton microscopy.

12- Line 492: Microtubule? Maybe you meant microfilament or F-actin? Please, verify.